

# 1  A Preliminary Assessment of the Impacts of

# 2  Multiple Temporal-scale Variations in Particulate

# 3  Matter on its Source Apportionment

Xing Peng[1], Jian Gao[2], Guoliang Shi[1*], Xurong Shi[1], Yanqi Huangfu[1], Jiayuan Liu[1],
Yuechong Zhang[2], Yinchang Feng[1**], Wei Wang[3], Ruoyu Ma[3], Cesunica E. Ivey[4], Yi
Deng[5]
[1]State Environmental Protection Key Laboratory of Urban Ambient Air Particulate
Matter Pollution Prevention and Control & Center for Urban Transport Emission
Research, College of Environmental Science and Engineering, Nankai University,
Tianjin 300350, China,
[2] Chinese Research Academy of Environmental Sciences, Beijing 100012, China,
[3] College of Software, Nankai University, Tianjin 300350, China,
[4]Department of Physics, University of Nevada Reno, Reno, Nevada, USA 89557,
[5]Earth and Atmospheric Sciences, Georgia Institute of Technology, Atlanta, GA 30332
**Correspondence to:**
**G.L. Shi, and Y.C. Feng,**
**nksgl@nankai.edu.cn;**
**fengyc@nankai.edu.cn**





**Abstract.** Time series of pollutant concentrations consist of variations at different time
scales that are attributable to many processes/sources (data noise, source intensities,
meteorological conditions, climate, etc.). Improving the knowledge of the impact of
multiple temporal-scale components on pollutant variations and pollution levels can
provide useful information for suitable mitigation strategies for pollutant control during
a high pollution episode. To investigate the source factors driving these variations, the
Kolmogorov-Zurbenko (KZ) filter was used to decompose the time series of $PM_{2.5}$
(particulate matter with an aerodynamic diameter less than 2.5 μm) and chemical
species into intra-day, diurnal, synoptic, and baseline temporal-scale components (TS
components). The synoptic TS component has the largest amplitude and relative
contributions (about 50%) to the total variance of $SO_4^{2-}$, $NH_4^+$, and OC concentrations.
The diurnal TS component has the largest relative contributions to the total variance of
$PM_{2.5}$, $NO_3^-$, EC, Ca, and Fe concentrations, ranging from 32% to 47%. To investigate
the source impacts on $PM_{2.5}$ from different TS components, four datasets RI (intra-day
removed), RD (diurnal removed), RS (synoptic removed), and RBL (baseline removed)
were created by respectively removing the intra-day, diurnal, synoptic, and baseline TS
component from the original datasets. Multilinear Engine 2 (ME-2) and/or principal
component analysis was applied to these four datasets as well as the original datasets
for source apportionment. ME-2 solutions using the original and RI dataset identify
crustal dust contributions. For the solutions from original, RI, RD, and RS datasets, the
total primary source impacts are close, ranging from 35.1 to 40.4 μg m$^{-3}$ during the
entire sampling period. For the secondary source impacts, solutions from the original,





RI and RD dataset give similar source impacts (about 30 µg m$^{-3}$), which were higher
than the impacts derived from the RS datasets (21.2 µg m$^{-3}$).
**Key words**: Multilinear Engine 2, Kolmogorov–Zurbenko filter, particulate matter,
temporal-scale components, source impact.





## 1 Introduction


Aerosol pollutants have become a major problem in recent years (Huang et al., 2014;
van Donkelaar et al., 2015), due to its negative influences on visibility, climate change
and human health (Langridge et al., 2012; Cheng et al., 2015; Butt et al., 2016; Ding et
al., 2017). Variations in aerosol concentrations and chemical species reflect influences
from multiple factors, such as local emissions sources and weather conditions, etc.
(Keim et al., 2005). Observed concentrations of pollutants, in general, have
characteristic variations, which are influenced by data noise, source intensities, short-
term fluctuations, source seasonal variation, meteorological condition, climate, policy,
and economic conditions (Milanchus et al., 1998; Wise and Comrie, 2005).
Online instruments can provide high time resolution data of particulate matter (PM)
and chemical species, and these instruments have been widely applied in the detection
of pollutants (Tchepel et al., 2010; Du et al., 2011; Zheng et al., 2015; Gao et al., 2016).
More and more studies on aerosol pollution have become dependent on high temporal
resolution observations due to their capabilities in revealing multiple temporal-scale
fluctuations of the aerosol concentrations that tend to arise from different physical,
chemical and dynamical processes. For example, during a high pollution period,
pollutant concentrations increase rapidly by several times over a short time period, and
such an increase tends to result from changing meteorological conditions. Hogrefe et
al. (2000) suggested that the time series of pollutant concentrations can be decomposed
into four components. The first component is the intra-day component with periods less



than 12 h and is typically linked to fast-acting, local emission sources and local-level
processes (Tchepel et al., 2010). The second is the diurnal component dominated by 12-
48 h periodicity. The third is the synoptic component mainly driven by 2-21 day
fluctuations in weather patterns and short-term fluctuations in emissions. The last
baseline component is related to the low-frequency fluctuations with periods greater
than 21 days, which might including seasonal or long-term scale variation in emissions,
climate, policy, etc. (Rao et al., 1997; Wise and Comrie, 2005).

Pollution sources are the key drivers of aerosol pollution. Understanding source

impacts on aerosols is important for the control of air pollution (Zhao et al., 2017).
Factor analysis models are widely used for estimating source impacts. These models
include principal component analysis/multiple linear regression (PCA/MLR), Unmix,
positive matrix factorization (PMF), and Multilinear Engine (ME-2) (Thurston and
Spengler, 1985; Paatero and Tapper, 1994; Henry and Christensen, 2010; Yin et al.,
2015; Zong et al., 2016). Among these, ME-2 is a particularly useful tool and has been
widely used in source apportionment studies (Paatero, 1999; Amato et al., 2009; Peng
et al., 2016). Factor analysis models depend on the variation of chemical species in
aerosols (which reflects the temporal variation of sources) to extract source categories
and calculate their contributions. Therefore, multiple temporal-scale variations in the
raw online datasets associated with various factors (e.g., data noise and weather
fluctuations) can have significant impacts on the source apportionment results using
factor analysis models. This is the main motivation behind our analysis, i.e., to
decompose the raw online datasets into multiple temporal-scale components and then



to estimate the influences of inclusion/exclusion of a specific temporal component on
the final apportionment results.

The Kolmogorov–Zurbenko (KZ) filter used in our study for extraction of a

specific temporal-scale component (TS component hereafter) is a low-pass filter that
has been widely used for decomposing temporal variations in $O_3$, PM, and chemical
species (Rao et al., 1997; Hogrefe et al., 2000, 2006; Wise and Comrie, 2005; Tchepel
et al., 2010). Hogrefe et al. (2006) reported that the synoptic component associated with
synoptic scale weather fluctuations has the largest relative contribution to the total
variance of hourly $PM_{2.5}$ concentrations, and the relative contributions of other
components to total $PM_{2.5}$ mass concentrations varies by chemical species in $PM_{2.5}$. In
addition, the noise of data might impact the analysis, and efforts have been made to
remove the noise (Kuebler et al., 2001; Tchepel et al., 2010; Henneman et al., 2015).
Our earlier studies demonstrated that the time resolution of the data could influence the
source apportionment results (Peng et al., 2016). Tchepel et al, (2010) used the KZ filter
to remove noise in the PM data, fed filtered PM data into air quality models and showed
that the model performance improved. In this study, we make a preliminary assessment
of the impacts of multiple temporal-scale variations in PM data on source
apportionment with $PM_{2.5}$ (PM with an aerodynamic diameter less than 2.5 μm) and its
chemical species observed in Beijing, China. Wavelet analysis was first used to evaluate
the periodicities of the PM and chemical species concentrations. The time series data
of the PM and chemical species were then decomposed into multiple TS components
using the KZ filter. Several new datasets were created by removing individual TS





components from the original receptor datasets. ME-2 or PCA analysis was conducted
on the original and new datasets to assess the impacts of excluding a specific TS
component on the final source apportionment results. We aim to determine what
processes/sources are responsible for the main variation characteristics and overall
pollution levels in this specific dataset. We also aim to determine what the implications
of our results are for source apportionment analyses conducted with data from different
geographical locations and under various weather/climate conditions.
**2    Methods**
**2.1 Sampling**
Ambient particles were collected in Beijing from 22 July 2014 to 12 August 2014 at
CRAES (Chinese Research Academy of Environmental Sciences) in this research. And
concentrations of $PM_{2.5}$, inorganic ions, OC/EC and heavy metals were measured by β-
ray monitor, model ADI 2080 online analyzer (MARGA, Applikon Analytical B.V.,
The Netherlands), OC/EC analyzer (Sunset Laboratory Inc, USA) and the Xact 625
automated multi-metals monitor (Copper USA), respectively, at 1 h time resolution.
Twenty-three chemical species were selected for analysis, including $NH_4^+$, $Na^+$, $Mg^{2+}$,
$Cl^-$, $NO_3^-$, $SO_4^{2-}$, K, Ca, Cr, Mn, Fe, Ni, Cu, Zn, As, Se, Ag, Cd, Ba, Hg, Pb, OC and
EC. The principles of these instruments and QA/QC are described in detail by Gao et
al. (2016).
**2.2 Source Impact Model**
ME-2, a general factor analysis model developed by Paatero (1999), was applied to



estimate the impacts of source categories at a location of interest. It is a general solver
of widely different multilinear and quasi-multilinear problems (Ramadan et al., 2003)
with the ability to deal with models consisting of a sum of products of unknowns.
Instead of being restricted to a specific structure, ME-2 is defined in a "script file" that
is written in a special-purpose programming language. It has efficient performance as
it runs in DOS, which made it faster than those models with graphical interfaces
(Ramadan et al., 2003). ME-2 decomposes original matrix $X_{(m \times n)}$ into source impact
matrix $G_{(m \times p)}$ and source chemical species (source profile) matrix $F_{(p \times n)}$, as follow:
$$X_{(m \times n)} = G_{(m \times p)} F_{(p \times n)} + E_{(m \times n)} \tag{1}$$
Variable $X_{(m \times n)}$ is the chemical species concentrations (unit: μg m$^{-3}$) in PM$_{2.5}$ that
are observed at the receptor site; $E_{(m \times n)}$ is the residual matrix; $m$ and $n$ are the
sample size and chemical species number, respectively; and $p$ is the number of sources.
The basic principle of ME-2 also can be expressed as follow:
$$x_{ij} = \sum_{k=1}^{p} g_{ik} f_{kj} + e_{ij} \quad \text{i} = 1, 2, \ldots, \text{m} \quad \text{j} = 1, 2, \ldots, \text{n} \tag{2}$$
where $x_{ij}$ is the element in matrix $X_{(m \times n)}$, which is the measured concentration of
the $j^{th}$ specie in the $i^{th}$ sample (μg m$^{-3}$); $g_{ik}$ is the element in matrix $G_{(m \times p)}$ and
is the impact of the $k^{th}$ source on the $i^{th}$ sample; $f_{kj}$ is the element of matrix of
$F_{(p \times n)}$ and is the concentration of the $j^{th}$ specie in the $k^{th}$ source (source profile);
and $e_{ij}$ is the element in the residual matrix (Hopke, 2003).
In ME-2, a priori information (e.g. chemical profiles and ratios) can be
incorporated as a target to be approximately accomplished. The prior information must



be handled in form of auxiliary equations (Paatero, 1999). Auxiliary equations are
included as additional terms $Q_{\partial ux}$ in an enhanced object function $Q_{enh}$ (Amato et al.,
2009; Amato and Hopke, 2012), the equation can be written as follows:
$Q_{enh} = Q_{main} + Q_{aux}$ (3)
The term $Q_{main}$ is described as follows:
$Q_{main} = \sum_{i=1}^{m} \sum_{j=1}^{n} (e_{ij}/\sigma_{ij})^2$ (4)
where $\sigma_{ij}$ is the uncertainty in the $j^{th}$ species for the $i^{th}$ sample; $e_{ij}$ has the same
meaning as is described in Eq.(2).

One of the simplest forms of the auxiliary equation is the "pulling equation"

(Paatero and Hopke, 2009), consisting of pulling $f_{kj}$ (for instance) toward the specific
target value $a_{kj}$:
$Q_{aux} = \frac{(f_{kj} - a_{kj})^2}{\sigma_{kj}^{aux2}}$ (5)
where $\sigma_{kj}^{aux}$ is the uncertainty connected to the pulling equation or softness of the pull;
and $f_{kj}$ is the element of factor loading. The task of ME-2 is to calculate a minimum
$Q_{enh}$ value or balance the minimization of the values $Q_{main}$ and $Q_{aux}$ in the
iterative process (Paatero and Hopke, 2009).

For ME-2, it requires that every element in the input dataset (matrix X) be a non-

negative value. Some datasets removing TS component in this work have negative
values and were analyzed using PCA instead of ME-2 due to this non-negative
requirement. PCA is a useful method to qualitatively identify the pollutant sources. It
reduces the number of original variables and generates a set of new variables (or called
principal components) that are ordered by the contribution to the total variance in the





original data.
**2.3 Temporal Scale Analysis**
The KZ filter is a widely applied filtering technique due to its powerful separation
characteristics, simplicity, and ability to handle missing data (Rao et al., 1997; Hogrefe
et al., 2006). The principle of KZ filter is described as follow:
$y_t = \frac{1}{m}\sum_{s=-(m-1)/2}^{(m-1)/2} x_{(t+s)}$                               (6)

$m$ is the length of the moving average window, which is an odd number; $x_{(t+s)}$

is the $(t+s)^{th}$ original value, $y_t$ is the average value. Then the $y_t$ as the input data
and calculate according to Eq. (6). After k times (number of iterations) calculation, $y_t^{(k)}$
is expressed as:
$y_t^{(k)} = KZ_{m,k}(X)$                               (7)

$y_t^{(k)}$ is removed the variations that frequency lower than $w$ (cutoff frequency).

$k$ is the number of iterations. Before conducting the KZ filter, the data is log
transformed for variance stabilization (Hogrefe et al., 2000). The separation point $w$,
between the high-frequency and low-frequency component, is a function of the filter
parameters m and k (Rao et al., 1997). The equation can be written as follows:
$w \approx \frac{\sqrt{6}}{\pi}\sqrt{\frac{1-(1/2)^{1/2k}}{m^2-(1/2)^{1/2k}}}$                               (8)

Selecting proper filter parameters m and k could remove the temporal component

at a specific frequency from the original dataset.

To select the appropriate parameters for the KZ filter, the wavelet analysis method

analyzed the $PM_{2.5}$ and chemical species' periodicities before decomposing their





concentrations time series. The results of wavelet analysis suggested that the
periodicities of PM$_{2.5}$ and chemical species are mainly 4-8 h (<12 h), 16-32 h, and 128-
256 h (6-10 day) (Figure S1, see the Supporting Information), which was similar to the
results reported by Hogrefe et al. (2000). This work referred to the KZ filter parameters
reported by the Hogrefe et al. (2000) study, and decomposed the PM$_{2.5}$ and chemical
species hourly concentrations into intra-day (time period less than 12 h), diurnal (12-
24 h), and synoptic (2-21 days) TS components to evaluate the influence of TS
components on their temporal variability. The formulas of the different TS components
are as follow:
$X_{(intra-day)} = X_{(Original)} - e^{KZ_{3,3}\{ln[X_{(Original)}]\}}$                                                    (9)
$X_{(diurnal)} = e^{KZ_{3,3}\{ln[X_{(Original)}]\}} - e^{KZ_{13,5}\{ln[X_{(Original)}]\}}$                           (10)
$X_{(synoptic)} = e^{KZ_{13,5}\{ln[X_{(Original)}]\}} - e^{KZ_{103,5}\{ln[X_{(Original)}]\}}$                       (11)
$X_{(baseline)} = X_{(Original)} - X_{(intra-day)} - X_{(diurnal)} - X_{(synoptic)}$                               (12)

$X_{(Original)}$ (original dataset) is the measured concentrations dataset including

PM$_{2.5}$ and chemical species (µg m$^{-3}$); $X_{(intra-day)}$, $X_{(diurnal)}$, $X_{(synoptic)}$, and
$X_{(baseline)}$ are concentration datasets of the intra-day, diurnal, synoptic and baseline
components, respectively (µg m$^{-3}$). The subscript numbers of KZ are descriptive
parameters. For example, the first "3" in KZ$_{3,3}$ is the length of the moving average
window, and the second "3" is the iteration time.
**2.4 TS Component Removed Datasets**
To exam the impact of the four TS components on the source impacts, datasets without



the TS component influence were created. Four datasets were created by respectively
removing the intra-day, diurnal, synoptic, and baseline TS component from the original
datasets, as follow:
$X_{(intra-day\ removed)} = e^{\text{KZ}_{3,3}\{ln[X_{(Original)}]\}}$ (13)
$X_{(diurnal\ removed)} = X_{(Original)} - X_{(diurnal)}$ (14)
$X_{(synoptic\ removed)} = X_{(Original)} - X_{(synoptic)}$ (15)
$X_{(baseline\ removed)} = X_{(Original)} - X_{(baseline)}$ (16)

$X_{(intra-day\ removed)}$ (RI dataset) is the concentration dataset with the intra-day

TS component removed from the original dataset ($\mu g\ m^{-3}$) and it contains the diurnal,
synoptic, and baseline TS components. $X_{(diurnal\ removed)}$ (RD dataset),
$X_{(synoptic\ removed)}$ (RS dataset), and $X_{(baseline\ removed)}$ (RBL dataset) are the datasets
with the diurnal, synoptic, and baseline TS components singly removed from the
original dataset, respectively. As there were many negative values in the RBL dataset,
these data were analyzed by PCA to qualitatively identify the sources of $PM_{2.5}$. The
original, RI, RD, and RS datasets were run by ME-2 for the source apportionment, and
their results were compared. Also, few negative values (very low count) were replaced
with a value equal to half of the detection limits. The average absolute error (AAE, see
the SI) and correlation analysis were employed to compare the differences in the source
impacts between original dataset and datasets with removed TS components. AAE was
employed and is calculated as follows (Javitz et al., 1988):
$AAE_k = \frac{1}{n} \times \sum_{i=1}^{n} \frac{|E_{ik} - T_{ik}|}{T_{ik}} \times 100$ (17)
where, $AAE_k$ is the $AAE$ value for the $k^{th}$ species and $n$ is the number of samples.
$E_{ik}$ is the concentration (μg m$^{-3}$) of the $k^{th}$ species for the $i^{th}$ sample from the RI,
RD, or RS datasets. $T_{ik}$ is the concentration (μg m$^{-3}$) of the $k^{th}$ species for the $i^{th}$
sample from the original dataset. The larger AAE value and the lower correlation
coefficients (r) indicate a larger difference in source impacts between the original
dataset and modified datasets, suggesting that the corresponding TS component has a
larger influence on the observed concentrations.
**3 Result and discussion**
**3.1 TS Component Influence on Concentrations**
The influence of each TS component on the pollutant concentration variation and the
concentration levels were investigated. The original dataset was decomposed into intra-
day, diurnal, synoptic, and baseline TS components by using the KZ filter (Figure 1).
The variation analysis was then employed to study each TS component contribution to
the total variance of PM$_{2.5}$ and the chemical species concentrations (Table 1). We placed
emphasis on investigating PM$_{2.5}$ and the source markers (e.g. SO$_4^{2-}$, Ca, OC, etc.),
because the variation of those markers can reflect the source emission pattern to some
extent.

The sample size of the four TS components was less than original dataset, because

the KZ filter was iterated with a moving average with a specified length and resulted in
missing head and tail data of the original data. The same period (from 24 July 2014 to
10 August 2014) of the original datasets with the same size were selected for





comparison and analysis. Among all the species, $PM_{2.5}$ and $NO_3^-$ had similar trends: the
diurnal and synoptic TS components had larger amplitudes and higher relative
contributions to the total variance of $PM_{2.5}$ (diurnal: 36%, synoptic: 32%) and $NO_3^-$
(diurnal: 36%, synoptic: 32%) than the intra-day and baseline TS components. $SO_4^{2-}$
and $NH_4^+$ showed similar variability: synoptic TS component had the largest amplitude
and had the largest relative contributions to the total variance of $SO_4^{2-}$ (48%) and $NH_4^+$
(54%) concentrations, followed by the baseline, diurnal and intra-day TS components.
OC was relatively different from the species mentioned above. For OC, the relative
contribution of the synoptic TS component was the largest (56%), followed by diurnal
(23%), baseline (12%) and intra-day TS components (9%). Species from primary
emission sources (such as EC, Ca, Fe, etc.) showed different patterns, compared with
the secondary species ($NO_3^-$, $SO_4^{2-}$, $NH_4^+$) discussed above. For EC and Ca, the diurnal
TS component had the largest relative contribution to the total variance of
concentrations, accounting for 47% and 45%, respectively. The synoptic (28%) and
intra-day (40%) TS component was the second largest contributor to the total variance
of EC and Ca concentrations, respectively. For Fe, diurnal and synoptic TS components
had larger amplitudes and higher relative contributions to the total variance than intra-
day and baseline TS components. For other elements, diurnal or intra-day TS
components had the largest amplitudes and were the larger contributors to the total
variance of the concentrations. Secondary organic carbon (SOC) also has been
estimated using the OC/EC ratio (see Supporting Information), and the influence of TS
component on the SOC was investigated (Table 1). The average concentration of SOC





was $5.7 \pm 3.1$ μg m$^{-3}$ in this work. For SOC, the diurnal and synoptic TS components
had larger amplitudes (Figure S2) and higher relative contributions to the total variance
of PM$_{2.5}$ (diurnal: 20%, synoptic: 62%). For species showing different TS component
contributions, the cause was external influencing factors. For example, variability in
primary species (such as Ca, EC) concentrations was mainly caused by local emission
patterns and meteorological diffusion(van Pinxteren et al., 2009); secondary species
were mainly influenced by chemical reaction (photochemical, liquid phase or
heterogeneous reaction) and meteorological conditions (Buzcu et al., 2006; Jung et al.,
2010, Martin et al., 2014). Therefore, species with similar TS component contributions
trends may have similar sources or influencing factors.

To investigate the influence of the TS components on concentration levels, partial

statistical analysis and AAE analysis were performed on the PM$_{2.5}$ and source markers
(NO$_3^-$, SO$_4^{2-}$, NH$_4^+$, Ca, Fe, OC, and EC) from five ambient datasets (including the
original, RI, RD, RS, and RBL datasets). The results are shown in Figure 2 and Table
S1. For ions, elements, OC/EC and PM$_{2.5}$, the larger gap in AAE value between the
concentrations of RBL and original dataset means a larger difference between them,
suggesting that the baseline TS component was the largest contributor to the average
concentrations of PM$_{2.5}$ and chemical species. The synoptic TS component also had a
relatively high contribution to the average concentrations of NO$_3^-$, SO$_4^{2-}$, and NH$_4^+$.
The average concentrations of the three ions of the RS dataset were obviously lower
and had large AAE values, compared with the results of the original dataset. For PM$_{2.5}$
and seven species, the correlation coefficients for the original dataset and RI, RD, RS,





and RBL datasets are displayed in Table S2. The lowest correlation coefficients were
obtained for the RBL TS components and the original data.
Overall, baseline TS components dominating the average concentrations of $PM_{2.5}$
and chemical species might imply that pollutant emissions and other long-term
fluctuation factors mainly determined the pollutants level in Beijing, from 22 July 2014
to 12 August 2014. When synoptic, diurnal, and intra-day TS components mainly
influenced the variation of $PM_{2.5}$ and chemical species, this suggests that the short-term
fluctuation (e.g. noise, weather etc.) dominantly determined the variation of pollutants.
**3.2 Source Impacts on $PM_{2.5}$ Concentrations**
Four datasets, including the original, RI, RD and RS datasets, were respectively
introduced into ME-2 to identify the sources of $PM_{2.5}$. The RBL was analyzed by PCA,
as several negative values were in this dataset (the ME-2 model only allows
nonnegative input values). The source apportionment results were explored, including
source profiles and source impacts, to investigate the source impacts on the $PM_{2.5}$
concentrations under the influence of different TS components.
For all four datasets, 3 to 7 factors were tested to determine the optimal number of
factors (source categories). There are some criteria for choosing the appropriate number
of factors, including the Q values, physical meaningfulness of the factor profiles, the
reasonableness of source impacts, and goodness of fit for $PM_{2.5}$ and chemical species
concentrations. After testing, four source categories were identified using ME-2 from
the original, RD, and RS datasets; five sources were obtained from the RI dataset. When





the calculated Q value close to the theoretical Q (Q_{the}), the corresponding results might
be acceptable (Hopke, 2003). The Q values of each solution calculated by ME-2 are
displayed in Table S3 and were close to the theoretical Q, further suggesting that the
results are acceptable.
Performance of ME-2 was evaluated by analyzing the goodness of fit for the
modeled and measured PM$_{2.5}$ and chemical species mass concentrations (slope, r).
Figure S3 illustrates the slope and r results, respectively. For PM$_{2.5}$, the slopes (ranging
from 1.0 to 1.1) and r (ranging from 0.8 to 1) were close to 1, suggesting the good
performance of ME-2 obtained for the five runs. For original dataset run, 13 out of 23
chemical species (e.g. SO$_4^{2-}$, OC, EC, Fe,) obtained slop values ranged from 0.80 to
1.20, and r of the corresponding species were varied from 0.60 to 0.96. Other species
(e.g. As, Cr, Se) obtained high slop values (larger than 1.20) and relative low r (ranged
0.01 to 0.84), indicating the poor precision of modeled results for those species.
Performance of solutions from RI and RD datasets are better than the solution from
original dataset, due to more species obtained good slops and r values (close to 1). For
RS run, slop values of chemical species range from 0.94 to 1.45, and values of six
chemical species larger than 1.20. The correlation coefficients are ranged from -0.06 to
0.95, which are similar with the results from the original dataset. The precision of
results from BL dataset is the best than the other four runs, because slop and r values
(larger than 0.89) of all species are close to 1. Summarily, receptor data filtering by KZ
filter approach can improve the performance of model results obtained by ME-2.
For the original dataset, four factor profiles were obtained (Figure 3). Factor 1 was
characterized by Ca and Fe, which is linked to crustal dust (Pant and Harrison, 2012).
Factor 2 had OC and EC, which are markers of vehicle emissions (Ramadan et al.,
2003). Factor 3 was identified as coal combustion due to high loadings of OC, EC and
Ca (Ramadan et al., 2003). Factor 4 was secondary formation due to elevated $SO_4^{2-}$,
$NO_3^-$ and $NH_4^+$ (Pant and Harrison, 2012). According to the previous studies (Yu et al.,
2013), coal combustion, secondary formation, vehicle emissions and crustal dust were
the dominating sources of $PM_{2.5}$ in Beijing. Five sources were obtained from the RI
dataset, including coal combustion, crustal dust, secondary formation, secondary nitrate
and vehicle emissions (Figure 3). For the RD and RS datasets, coal combustion,
secondary formation, secondary nitrate and vehicle emissions were identified (Figure
3), and crustal dust was not identified and was mixed with vehicle emissions from the
two datasets. It was an expected result that ME-2 failed to identify the crustal dust
source for the RD dataset. Because ME-2 extracts factors based on the chemical species
variation pattern (the marker species variation can reflect the source emission pattern
over the time), crustal dust markers (Ca and Fe) lost much variance and could not reflect
the expected pattern after removing the diurnal TS component (the largest contributor
to the total variance of Ca and Fe) (Table 1). As for the solution for RS dataset, the
sulfate source (not the secondary formation) and the nitrate source were distinguished
as different factors (Figure 3). For the solutions of the original, RI, and RD datasets,
secondary formation of sulfate and nitrate source were mixed together and extracted as
one factor. We found that after removing the similar part of the variance (the
information filtered by KZ filter) of sulfate and nitrate, the difference between sulfate



and nitrate variation trend was more obvious, so these two sources can be distinguished
by ME-2. It can be confirmed that the correlation coefficient between $NO_3^-$ and $SO_4^{2-}$
was highest (0.86) for the synoptic TS component compared with other TS components
(0.41, 0.28, and 0.82 for intra-day, diurnal, and baseline TS components, respectively).
The correlation was lowest (0.49) for the RS dataset compared with other datasets (0.68,
0.72, and 0.85 in the original, RI, and RD datasets, respectively).
The PCA results of the RBL dataset are listed in Table S4. Seven factors were
extracted and accounted for 80.9% of the total variance, which had corresponding
eigenvalues larger than 1 (can be considered as the potential sources). Factor 1 (19.6%
of the variance) had high loadings for heavy metals, such as As, Se, Pb, etc. Factor 2
had high loadings for Ca and Ba, and relatively high loadings for Mn, Fe, and EC. This
factor might be associated with crustal dust and had a 14.3% contribution to the
variance (Pant and Harrison, 2012). High loadings were observed for $SO_4^{2-}$, $NO_3^-$, and
$NH_4^+$ in factor 3, which is associated with secondary formation (Pant and Harrison,
2012). Factor 4 had relatively high loadings for Cu and $Cl^-$, and factor 5 and 6 were
characterized by heavy metals. After removing the baseline dataset, heavy metals and
crustal dust were the dominant sources of $PM_{2.5}$. According to the results of the RBL
dataset, we found the evidence that intra-day and diurnal TS components had larger
relative contributions to the total variance of element (heavy metals) concentrations, as
these elements are mainly emitted from primary sources.
Because there were different sample sizes for the four datasets, we selected the
same period of results (from 24 July 2014 to 10 August 2014) to study the influence of



TS components on the source variation (Figure 4). The time series of source impacts
from the RI, RD, and RS datasets were respectively used for correlation analysis with
the corresponding results from the original dataset (Table S5). Vehicle emissions
solutions from the RI ($r = 0.45$) and RD ($r = 0.51$) datasets had a higher correlation than
the RS dataset ($r = 0.25$), suggesting that intra-day and diurnal TS components had
stronger influences on the source pattern (variation) of vehicle emissions. For coal
combustion, results from the RI, RD, and RS datasets had similar correlation
coefficients (ranging from 0.74 to 0.82). The sulfate source was identified from the RS
dataset (Figure 3), however, the sulfate source had the lowest correlation with
secondary formation (Table S5) solutions from the original dataset. The correlation
analysis of the nitrate source produced similar results, where the lowest correlation
coefficient occurred between solutions from the RS dataset and original dataset,
suggesting that secondary source impact variation is dominantly affected by synoptic
scale influences.
To further investigate source impacts on $PM_{2.5}$ from different TS components, we
discussed the average impacts of individual source categories on $PM_{2.5}$ from the
datasets with removed TS components (Table 2).
Vehicle emissions, crustal dust, and coal combustion were combined together for
the analysis (called as TPS: total primary sources), because crustal dust was mixed with
vehicle emissions and coal dust for the RD and RS datasets, as mentioned above.
Secondary formation and nitrate source were also plus together for the discussion
(called as TSS: total secondary sources). To better explore the influence of TS



components, source impacts during the entire sampling period and pollution period
were investigated separately. For the entire sampling period, the impacts of TPS
obtained from the original, RI, RD, and RS datasets were similar to each other, ranging
from 35.1 to 40.4 μg m$^{-3}$. This was an expected result because the intra-day, diurnal,
and synoptic TS components had small influence on the concentrations levels of
primary source markers (OC, EC, elements), as shown in Figure 2. The TSS solutions
from the original, RI, and RD datasets exhibited similar source impacts, accounting for
about 30 μg m$^{-3}$, which was higher than the solution from the RS dataset (21.2 μg m$^{-3}$).
The synoptic TS component had impact on the $SO_4^{2-}$, $NO_3^-$ and $NH_4^+$ concentrations,
and removing this TS component may have resulted in lower impacts of the secondary
sources.
During the pollution period (from 30 July to 4 August 2014, gray shadow shown
in Figure 1), the highest concentration of $PM_{2.5}$ was up to 183.7 μg m$^{-3}$ at 1:00 am on
31 July 2014, with an average concentration of 85.5 μg m$^{-3}$. The TPS impacts derived
from the original, RI, RD, and RS datasets were relatively stable, ranging from 30.3 μg
m$^{-3}$ to 37.4 μg m$^{-3}$ (Table 2). The TSS impacts from the RS dataset (29.3 μg m$^{-3}$) were
lower than the solutions from the original, RI, and RD runs (about 51 μg m$^{-3}$). The
synoptic TS component increased the $NO_3^-$, $SO_4^{2-}$, and $NH_4^+$ concentrations (Figure 1),
accounting for 58%, 57%, and 50% of their original average concentrations,
respectively. This implies that the synoptic TS component had a larger impact on the
secondary source than the primary source during the pollution period. Liu et al. (2017)
reported that some haze episodes in North China Plain (including Tianjin) resulted from



elevated relative humidity (RH) and stagnant weather conditions. The study proposed
an inorganic aerosol formation mechanism for which the elevated RH and the inorganic
fraction increased the aerosol liquid water content (LWC), then the liquid particles
would uptake pollutants to form the aerosols. In this work, elevated RH and reduced
wind speeds have been observed during the pollution period (Figure S4). To further
confirm the assumption, the aerosol LWC was estimated using ISORROPIA II model
(Guo et al., 2015). The LWC and total ions ($NO_3^-$+$SO_4^{2-}$+$NH_4^+$) depending on RH are
shown in Figure S5. Under similar RH conditions, the total ions (ranged from 93 to 121
$\mu g\ m^{-3}$) and LWC concentrations (ranged from 48 to 513 $\mu g\ m^{-3}$) during the pollution
period were higher than the corresponding values (total ions concentrations: 22 to 38
$\mu g\ m^{-3}$; LWC: concentrations 12 to 108 $\mu g\ m^{-3}$) during the non-pollution, suggesting
high total ion and LWC concentrations. When the RH lower than 90% during the
pollution period, the total ions concentrations remained relatively stable (about 100$\mu g$
$m^{-3}$), while the LWC concentrations increased. When the RH was higher than 90%, the
total ions and LWC concentrations increased to 120 and 513 $\mu g\ m^{-3}$ during the pollution
period, respectively, suggesting the drastically increasing in LWC concentrations may
have led to the elevated ions concentrations. Therefore, the stagnant weather conditions
and the elevated RH may increase the inorganic concentrations during the pollution
period in this work.

Overall, removing the different TS components had little influence on primary

source impact levels, suggesting that primary source impact levels were mainly
influenced by the source emissions. The secondary source impact levels were mainly




influenced by synoptic influences and source emissions.

The BL dataset may be linked with source emissions, source seasonal variance,

and long-term meteorological fluctuations. To study the source impacts from the
baseline TS component, ME-2 was applied to the baseline dataset. Four sources were
identified, including the nitrate source, secondary formation, coal combustion, and
vehicle emissions (Figure S6). During the entire sampling period (Table 3), the average
TPS and TSS impacts on $PM_{2.5}$ mass concentrations were 29.9 µg m$^{-3}$ (57%) and 22.8
µg m$^{-3}$ (43%) respectively. The average impacts of TPS and TSS during the pollution
period were higher than the corresponding average impacts during the entire sampling
period, which were 35.6 µg m$^{-3}$ and 26.0 µg m$^{-3}$, respectively (Table 3). TPS and TSS
obtained same impact percentages from the entire period and pollution period,
accounting for 58% and 42% of $PM_{2.5}$ mass concentrations, respectively. The time
series of TPS, TSS, and $PM_{2.5}$ are shown in Figure S7 and suggest that the periodicities
of TPS and TSS were not synchronized. In this work, the peak of the BL TS component
of $PM_{2.5}$ was obtained when the both the TPS and the TSS impact levels were high.
**4 Conclusions**
In this work, KZ filter was applied to decompose the time series of $PM_{2.5}$ and chemical
species concentrations collected in Beijing into intra-day, diurnal, synoptic, and
baseline temporal-scale (TS) components. This work investigated the factors driving
these variations, and influencing factors were found to vary with species. The intra-day
and diurnal TS components mainly influence the fluctuation of elements concentrations



(e.g. Ca, Cr, Mn, etc); diurnal and synoptic TS components mainly impacted the
fluctuation of $PM_{2.5}$, $NO_3^-$, EC, and OC concentrations; baseline and synoptic TS
components were the main factors contributing to $SO_4^{2-}$ and $NH_4^+$ variance. For the
$PM_{2.5}$ and all chemical species concentration levels, the baseline TS component was the
dominant factor.
To study the influence of different TS components on the source impacts on $PM_{2.5}$,
four datasets (RI, RD, RS, and RB) were created by removing one individual TS
component from the original dataset each time. The original and the four modified
datasets were analyzed by ME-2 and/or PCA, and the source apportionment results
were compared. We found that removing some TS components affected the source
identification. Four sources were obtained from the original and RI analyses, including
crustal dust, vehicle emissions, coal combustion, and secondary formation. Crustal dust
was not identified by ME-2 from the RD, RS, and BL datasets, possibly due to the fact
that much of the information regarding the markers of crustal dust (e.g. Ca) was lost
after removing the corresponding TS components. This suggests that the diurnal and
synoptic TS components of chemical species were important for identifying the crustal
dust source.
For the solutions from the original, RI, RD, and RS datasets, TPS (including
crustal dust, vehicle emissions, and coal combustion) were similar to each other,
implying that intra-day, diurnal or synoptic TS components had little influence on the
TPS impact levels. The TSS (secondary formation and nitrate source) from the original,
RI, and RD datasets obtained similar source impact levels; while TSS impact from the



RS dataset was lower than other three results, suggesting that TSS was mainly
influenced by the synoptic TS component and source emissions. Performance of four
ME-2 runs was evaluated by analyzing the goodness of fit for the modeled and
measured $PM_{2.5}$ and chemical species mass concentrations (slope, r). Receptor data
filtering intra-day TS components by KZ filter approach can improve the performance
of the model and produce reasonable source impact results, suggesting that filtering
noise from the instrument is useful to data analysis.

The major findings of this work are that during the whole sampling period and

pollution period, TPS impact levels were mainly influenced by source emissions, and
TSS impact levels were mainly influenced by synoptic scale weather fluctuations and
source emissions. The future work will focus on the mechanism through which synoptic
scale weather disturbances modulate the secondary species and sources.

*Data availability*. The data used in this study are available from the corresponding
author upon request (**nksgl@nankai.edu.cn; fengyc@nankai.edu.cn**).
*Competing interests*. The authors declare that they have no conflict of interest.

*Acknowledgments.* This study was supported by the National Natural Science
Foundation of China (No. 41775149, 91544226), the National Key Research and
Development Program of China (No. 2016YFC0208500, 2016YFC0208505), the
Tianjin science and technology Foundation (No. 16YFZCSF00260), the Tianjin
Natural Science Foundation (No. 17JCYBJC23000).



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





**Tables**
**Table 1.** Relative contributions (%) of the different TS components to the total variance of PM$_{2.5}$
and chemical species concentrations.

|  | NO$_3^-$ | SO$_4^{2-}$ | NH$_4^+$ | OC | EC | Ca | Fe | SOC | PM$_{2.5}$ |
|---|---|---|---|---|---|---|---|---|---|
| Intra-day (%) | 5 | 4 | 4 | 9 | 17 | **40** | 20 | 9 | 9 |
| Diurnal (%) | **36** | 18 | 17 | 23 | **47** | **45** | **32** | **20** | **36** |
| Synoptic (%) | **32** | **48** | **54** | **56** | 28 | 10 | **32** | **62** | **32** |
| Baseline (%) | 27 | 31 | 26 | 12 | 8 | 5 | 16 | 9 | 24 |






**Table 2.** Average source contributions to PM$_{2.5}$ (μg m$^{-3}$) estimated by ME-2 from Beijing for the
original, RI, RD, and RS datasets during the entire sampling period.

| | | Crustal dust | Vehicle emission | Coal combustion | TPS[a] | Secondary formation | Nitrate source | TSS[b] |
|---|---|---|---|---|---|---|---|---|
| During the entire sampling period | Original | 14.2 (20%) | 15.2 (22%) | 11 (16%) | **40.4 (58%)** | 29.4 (42%) | | **29.4 (42%)** |
| | RI | 8.6 (13%) | 12.7 (19%) | 18.5 (27%) | **39.9 (58%)** | 26.3 (38%) | 2.3 (3%) | **28.7 (42%)** |
| | RD | | 14.4 (22%) | 20.8 (32%) | **35.1 (54%)** | 26.2 (41%) | 3.2 (5%) | **29.4 (46%)** |
| | RS | | 19.2 (32%) | 19.8 (33%) | **39 (65%)** | 6.8 (11%) | 14.4 (24%) | **21.2 (35%)** |
| Pollution period | Original | 13.5 (16%) | 12.3 (14%) | 7.7 (9%) | **33.5 (39%)** | 52.1 (61%) | | **52.1 (61%)** |
| | RI | 6 (7%) | 10.1 (12%) | 17.9 (21%) | **34 (40%)** | 50.3 (59%) | 1.0 (1%) | **51.2 (60%)** |
| | RD | | 14.8 (18%) | 15.5 (19%) | **30.3 (37%)** | 48.9 (60%) | 2.5 (3%) | **51.4 (63%)** |
| | RS | | 24.4 (37%) | 13.1 (20%) | **37.4 (56%)** | 10.8 (16%) | 18.5 (28%) | **29.3 (44%)** |

[a]TPS is the total contributions of crustal dust, vehicle emissions, and coal combustion. [b]TSS is the
total contributions of secondary formation and nitrate source.





**Table 3.** Average source contributions to PM$_{2.5}$ (μg m$^{-3}$) estimated by ME-2 from the BL datasets.

| | Vehicle emission | Coal combustion | TPS[a] | Secondary formation | Nitrate source | TSS[b] |
|---|---|---|---|---|---|---|
| During the entire sampling period | 15.4 (29%) | 14.5 (28%) | **29.9 (57%)** | 12.0 (23%) | 10.8 (20%) | **22.8 (43%)** |
| Pollution period | 17.6 (29%) | 17.9 (29%) | **35.6 (58%)** | 22.4 (36%) | 3.6 (6%) | **26.0 (42%)** |

[a]TPS is the total contributions of crustal dust, vehicle emissions, and coal combustion. [b]TSS is the
total contributions of secondary formation and nitrate source.





## Figures

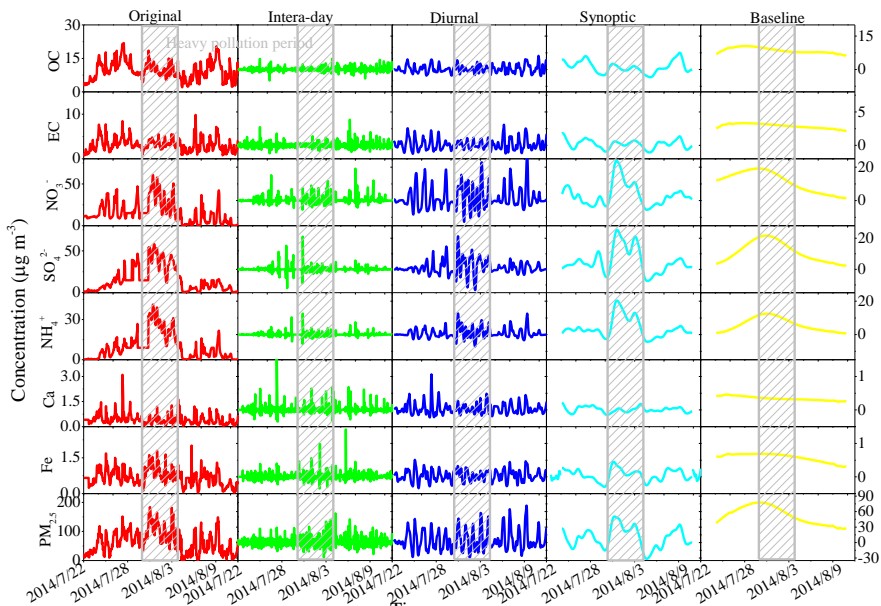

**Figure 1.** Variance of PM$_{2.5}$ and chemical species concentrations influenced by intra-day (time

period less than 12 h), diurnal (12-24 h), synoptic (2-21 days), and baseline (greater than 21 days)

temporal-scale (TS) components, for the period of 22 July 2014 to 13 Aug 2014 at Beijing, China.

The variation of species that originated from primary sources mainly were influenced by diurnal TS

components. The variation of ions and OC (partly from secondary formation) that originated from

secondary formation mainly were influenced by synoptic TS component. The vertical gray lines

demarcate the heavy pollution period.



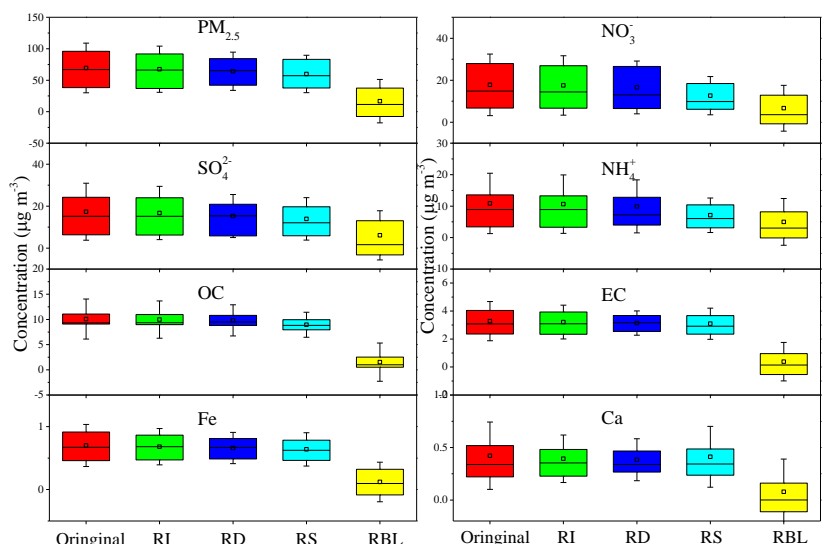

**Figure 2.** The influence of different TS components on the average concentrations of PM$_{2.5}$ and

chemical species. The baseline TS component dominated the PM$_{2.5}$ and chemical species average

concentrations. Presented are box plots of individual chemical species from the original, RI, RD,

RS, and RBL datasets. Cubes denote the average and dashes denote the median concentration. The

whiskers are the standard deviation. (RI: intra-day removed dataset, RD: diurnal removed dataset,

RS: synoptic removed dataset, RBL: baseline removed dataset).





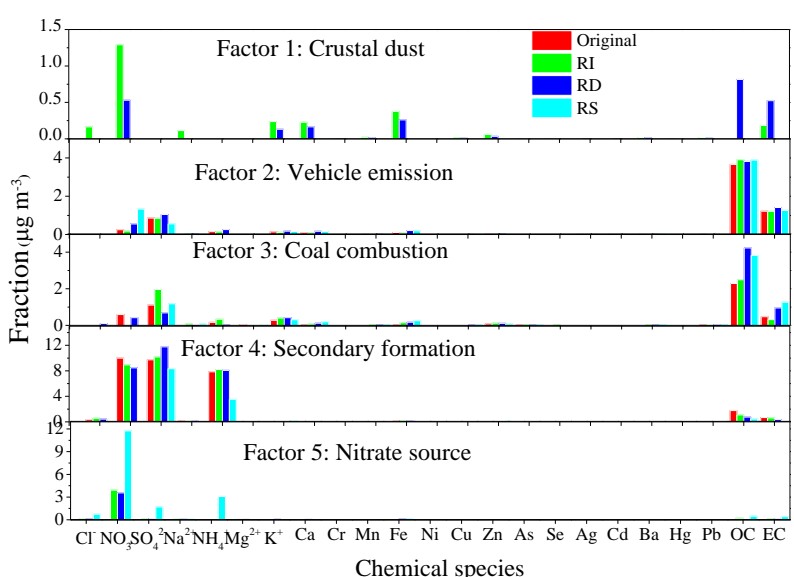

709

**Figure 3.** The influence of different TS components on source determination. Crustal dust was not

identified from the RD and RS datasets. Nitrate source and sulfate source were identified from the

RS dataset. Note: The factor 4 solution was generated from removing the synoptic dataset and

represents the sulfate source.





714

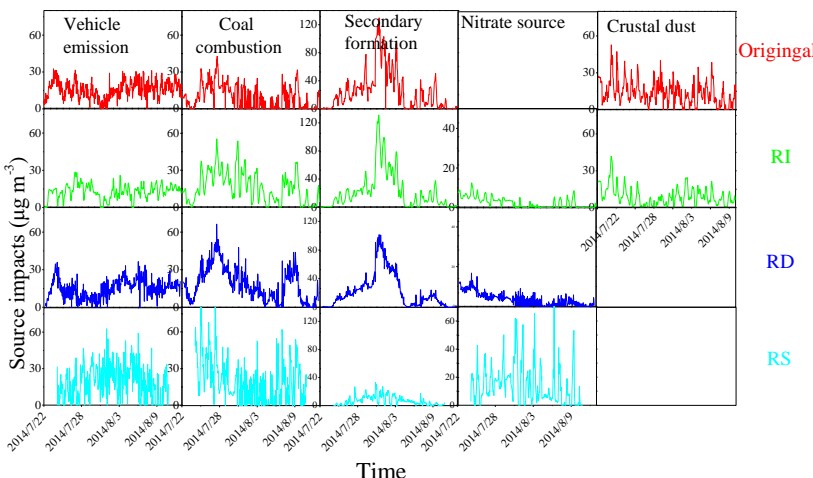

715

**Figure 4.** Source contributions to PM$_{2.5}$ for each source (vertical columns) and each TS component

(horizontal rows). The blanks mean that the source has not been identified.

718