# Peer review of "Discussion started: 7 March 2018"

_Atmospheric Chemistry and Physics, 2017_

## Referee Comment (RC1) · Anonymous Referee #3 · 9 Jun 2018

General comments: This study conducted a preliminary assessment of the impacts of multiple temporal-scale variations in PM data (using Kolmogorov-Zurbenko filter) on chemical species and source apportionment results, and tried to determine what processes/sources are responsible for the main variation characteristics. The method in this manuscript might be useful in the future PM source apportionment and air pollution studies. However, there are a few questions that are needed to be addressed before considering an acceptance of this work by Atmospheric Chemistry and Physics. 1. The authors should clarify the physical meaning of the four different TS components. 2. The authors should further clarify the different results from Figure 1 and Figure 2. For example, as shown in Figure 1, it seems that only variations of secondary inorganic

species (e.g., $SO_4^{2-}$, $NO_3^-$, $NH_4^+$) are more influenced by baseline TS component, and the relative influence by synoptic TS component were higher compared to baseline TS component. However, the results indicated by Figure 2 show that baseline TS component dominates the concentrations of PM2.5 and chemical species. 3. The different source profiles obtained from ME-2 look not quite good in the source apportionment results part. For example, chemical species profiles for coal combustion and vehicle emissions look quite similar both in Figure 3 and Figure S6; and there are high OC factions exist in the "Nitrate source". The authors should verify their source apportionment results and give more robust results and explanations. 4. The authors should carefully check their manuscript. There are some mistakes and inconsistences in the context and figures. Grammar modification is needed.

Specific comments: 1) Page 4 Line 55: change to "such as local emission sources and weather conditions" 2) Page 4 Line 78: change "key drivers" to "one of the key drivers"? 3) Page 9 Line 158-172: please clearly definite the abbreviations, such as Qmain, Qaux, akj 4) Page 10 Line 185: The authors should clarify what m values (the length of the moving average window) they used and what should be considered. 5) Page 10 Line 186: Grammar mistakes. Please revise the sentence "Then the yt as the input data and calculate according to Eq.(6)." 6) Page 10 Line 190: Grammar mistakes. Please revise the sentence "yt(k) is removed the variations that frequency lower than w (cutoff frequency)" 7) Page 11 Line 202: change "was" to "were" 8) Page 11 Line 202: Please give more descriptions on Figure S1. For example, what do the color bars and black lines represent? 9) Page 11: please clarify why different lengths of the moving average window (e.g., 3, 13, 103) were used in the X(intra-day), X(diurnal), and X(synoptic) calculation. And how was the w value determined? What about the criteria? 10) Page 14 Line 268: Remove the word "had". 11) Page 14 Line 280 and 282: Please show the data/figure for other elements in the table/figure in the manuscript or in the supplementary materials. 12) Page 15 Line 293: delete "s" in the word "contributions" 13) Page 15 Line 297: add "e.g." before "NO3-" 14) Figure 2: Please revise the sentence "Presented are box plots of individual chemical species

from the original, RI, RD, RS, and RBL datasets." 15) Table S3: Please definite the abbreviation "Qthe" 16) Figure S3: Please revise "BL" to "RBL" 17) Page 17 Line 337: Remove comma after "Fe" 18) Page 17 Line 346: It should be "RBL", but not "BL" 19) Page 17 Line 345-347": Please further explain why "the precision of results from RBL dataset is the best than other four runs" with "slop and r values (larger than 0.89 of all species close to 1)" 20) Page 18 Line 358-362: The context about results for RD and RS dataset are not consistent with those shown in Figure 3. Please check Figure 3 and the figure caption. 21) Figure S4: The vertical gray lines didn't correctly present the heavy pollution period selected "from 30 July to 4 August 2014" (see Line 427 on Page 21) 22) Table 3: Please clarify the data in the parentheses and outside the parentheses. 23) Page 23 Line 461: Please change "BL" to "RBL", and check other places. It seems the authors used "BL" incorrectly in the last part of the manuscript... 24) Page 23 Line 470: Please change "same" to "similar" 25) Page 24 in the conclusion part: Please use uniform and consistent abbreviations for "baseline removed datasets", e.g., "RBL", "RB", "BL".

Please also note the supplement to this comment:
https://www.atmos-chem-phys-discuss.net/acp-2017-997/acp-2017-997-RC1-supplement.pdf

---

## Referee Comment (RC2) · Anonymous Referee #4 · 24 Jul 2018

In this study, Kolmogorov-Zurbenko (KZ) filter was used to decompose the time series of PM2.5 and chemical species into intra-day, diurnal, synoptic, and baseline temporal-scale components (TS components), which might be helpful for a better understanding of source apportionment. However, I did not see good evaluation criteria to judge whether this relatively new approach is superior to the ME-2 or PMF without KZ. For the KZ method, different solution can be used, but it is still very questionable which solution is the best one. Therefore, I strongly suggest the authors extend the discussion to address my concern before a publication in ACP. In general, I think the author should use PMF instead of PCA for the source apportionment. In NCP, haze episode is much more frequent in winter than in summer, so the authors should use or include the winter

data set for a typical application for this new approach. I agreed with the other reviewer that the overall source apportionment result is not good enough to separate different sources. Therefore, seasonal variations should be included. I would like to see diurnal variations for traffic sources and other sources as well.

———————————————

---

## Referee Comment (RC3) · Anonymous Referee #5 · 27 Jul 2018

Review for: A Preliminary Assessment of the Impacts of Multiple Temporal-scale Variations in Particulate Matter on its Source Apportionment Atmos. Chem. Phys. Discuss., https://doi.org/10.5194/acp-2017-997

The authors have proposed to improve the identification of major sources of ambient PM2.5 by conducting source apportionment upon the spectral decomposition of hourly measurements of ambient PM2.5 and its major ionic and trace element components. I think the manuscript falls short of fulfilling the above purposes. Several issues arise after reviewing the manuscript:

a) The process of decomposing the original data in time series of different tempo-

ral variations does not ensure that each of them are strictly positive, as they should. This is problematic for the subsequent application of ME2 (or PMF). b) There is no quantification of uncertainties associated to each spectral decomposition. This is problematic for a correct application of ME2/PMF. Besides, the subtractions that appear in equations (9) − (16) increase the uncertainties of individual spectral components. This degrades input data quality, potentially hampering source identification. c) From equations (12) and (16) it follows that the RBL dataset is the sum of three spectral components. Nonetheless, the authors state that the above set has "many negative values" (line 233). This is a strange result and casts doubt on the suitability of the above decomposition for generating suitable input data for ME2/PMF. c) The dataset is too limited to validate the proposed methodology. Only two months of hourly data have been used. Monthly seasonality is absent in the analyses, and this is a serious flaw. Another limitation of this small datasets is the poor fitting of As, Cr and Se after applying ME2 to the original dataset (lines 338-340). These trace elements may come from intermittent (point) sources that arrive to the receptor site only when specific meteorological conditions are met. The data set is too small to resolve those sources. d) The authors acknowledge that the crustal dust source was not resolved in several data sets (RD, RS, BL) (lines 492-495). This contradicts their statement (on lines 506-509) that the proposed methodology improves source identification. See also c) above. e) Actually, the point in receptor modeling analyses is not removing the noise, but properly quantifying it, so noisy data are downweighted. f) The best input data for applying receptor models is one in which data variability is captured as much as possible, to sample all potential sources impacting a site (including intermittent sources); this also implies sampling for at least a year to include seasonality. The proposed methodology goes in the opposite direction: first, decomposing original data variability into several spectral components (each with lower variability) and, second, analyze each of them separately by receptor models.

Therefore, my recommendation is to reject the manuscript.

---

## Author Comment (AC1) · 5 Sep 2018

We want to thank the reviewers for their support of the paper and their helpful comments and suggestions which have helped to improve our manuscript. We provide a point by point response to review. Response to Anonymous Referee #1

**General comments:**

This study conducted a preliminary assessment of the impacts of multiple temporalscale variations in PM data (using Kolmogorov-Zurbenko filter) on chemical species and source apportionment results, and tried to determine what processes/sources are responsible for the main variation characteristics. The method in this manuscript might be useful in the future PM source apportionment and air pollution studies. However, there are a few questions that are needed to be addressed before considering acceptance of this work by Atmospheric Chemistry and Physics.

The authors should clarify the physical meaning of the four different TS components.
 Response: We have explained the physical meaning of the four TS components. The revised manuscript now have:

" $X_{(intra-day)}$ , is concentration dataset of the intra-day component that relates to fastacting, local emission sources, and local-level processes.  $X_{(diurnal)}$  is concentration dataset of the diurnal component that reflects the source diurnal variation.  $X_{(synoptic)}$ is concentration dataset of the synoptic component that is influenced by weather patterns and short-term fluctuations in emissions.  $X_{(baseline)}$  is concentration dataset of the baseline component that might link to seasonal or long-term scale variation in emissions, climate, policy, etc.".

2. The authors should further clarify the different results from Figure 1 and Figure 2. For example, as shown in Figure 1, it seems that only variations of secondary inorganic species (e.g.,  $SO_4^{2-}$ ,  $NO_3^{-}$ ,  $NH_4^+$ ) are more influenced by baseline TS component, and the relative influence by synoptic TS component was higher compared to baseline TS component. However, the results indicated by Figure 2 show that baseline TS component dominates the concentrations of PM2.5 and chemical species.

Response: According to Table 1, synoptic TS component is the largest contributor to the total variance of the three ions concentrations, followed by baseline TS component for SO42- and NH4+ and diurnal TS component for NO3- concentrations. Synoptic TS component had higher relative contributions to the total variance of secondary species compared to baseline TS component. However, Figure 1 shown the synoptic TS component not only increased the secondary species concentrations (>0) but decreased the values (<0) during the sampling campaign resulting in the small influence on the average concentrations of secondary species. Baseline TS component had higher concentration levels of secondary species although it had a little impact on the variations of secondary species. The baseline TS component has a small influence on the variations of species, including OC, EC, Ca, Fe. Figure 1 shown higher concentration levels of baseline TS component than the other three TS components, suggesting the baseline TS component is the bigger contributor to the species (OC, EC, Ca, Fe) average concentrations. Therefore, the results shown in Figures 1 and 2 were consistent that baseline TS components dominated the average concentrations of PM2.5 and chemical species.

|               | •                 |                                      |                       |    |    |           |    |           |                          |
|---------------|-------------------|--------------------------------------|-----------------------|----|----|-----------|----|-----------|--------------------------|
|               | NO 3 - | SO 4 2- | $\mathbf{NH}_{4^{+}}$ | OC | EC | Ca        | Fe | SOC       | PM 2.5 |
| Intra-day (%) | 5                 | 4                                    | 4                     | 9  | 17 | 40 | 20 | 9         | 9                        |
| Diurnal (%)   | 36                | 18                                   | 17                    | 23 | 47 | 45        | 32 | 20        | 36                       |
| Synoptic (%)  | 32                | 48                            | 54                    | 56 | 28 | 10        | 32 | 62 | 32                       |
| Baseline (%)  | 27                | 31                                   | 26                    | 12 | 8  | 5         | 16 | 9         | 24                       |

**Table 1.** Relative contributions (%) of the different TS components to the total variance of  $PM_{2.5}$  and chemical species concentrations.

---

## Author Comment (AC2) · 5 Sep 2018

We want to thank the reviewers for their support of the paper and their helpful comments and suggestions which have helped to improve our manuscript. We provide a point by point response to review.

**Response to Anonymous Referee #2**

In this study, Kolmogorov-Zurbenko (KZ) filter was used to decompose the time series of PM2.5 and chemical species into intra-day, diurnal, synoptic, and baseline temporal scale components (TS components), which might be helpful for a better understanding of source apportionment. However, I did not see good evaluation criteria to judge whether this relatively new approach is superior to the ME-2 or PMF without KZ. For the KZ method, different solution can be used, but it is still very questionable which solution is the best one. Therefore, I strongly suggest the authors extend the discussion to address my concern before a publication in ACP. In general, I think the author should use PMF instead of PCA for the source apportionment. In NCP, haze episode is much more frequent in winter than in summer, so the authors should use or include the winter data set for a typical application for this new approach. I agreed with the other reviewer that the overall source apportionment result is not good enough to separate different sources. Therefore, seasonal variations should be included. I would like to see diurnal variations for traffic sources and other sources as well.

Response: a) This work aims to the influence of different temporal-scale components on source contributions using Kolmogorov-Zurbenko (KZ) filter method and receptor model rather than establishes the new approach and compares it with PMF/ME2. We found that primary source impact levels mainly determined by baseline TS component (source emissions), and secondary source impact levels were mainly influenced by synoptic and baseline TS component (synoptic scale weather fluctuations and source emissions).

b) We reanalyzed a new dataset (about 5 months) that collected in Tianjin, China to strongly support our findings.

We collected dataset during winter in Tianjin, China and analyzed the dataset following the same process of Beijing dataset. Results from Tianjin datasets shown that diurnal and synoptic TS components had higher relative contributions to the total variance of OC, EC, PM$_{2.5}$, and most ions and elements than the intra-day and baseline TS components. Six source categories were identified using ME-2 from both original and RD datasets in Tianjin, including crustal dust, vehicle emissions, coal combustion, secondary formation, biomass burning & sea salt, and industrial source. While industrial source has not been identified from both RD and RS dataset that mixed with crustal dust. The TPS impacts calculated from the original, RI, RD, and RS datasets were similar (ranged from 33.8 to 38.2 μg m$^{-3}$), the TSS impacts derived from the original and RI datasets exhibited similar source impacts (about 33 μg m$^{-3}$), which was higher than the solution from the RD (27.3 μg m$^{-3}$) and RS dataset (23.0 μg m$^{-3}$). Both Beijing and Tianjin results suggested TPS impact levels were mainly influenced by baseline TS component, and TSS impact levels were mainly influenced by synoptic and baseline TS components. Seasonal variations of source impacts were investigated in Tianjin. A more significant decrease in secondary source impact during winter suggested that the synoptic TS component had a more significant impact on the secondary source during winter than during the fall in this work.

Detailed discussion as follow:

Long-Term observations were conducted in Tianjin, China, from October 03, 2017 to March 16, 2018. PM$_{2.5}$, inorganic ions, OC/EC, and heavy metals were measured by

β-ray monitor, an ambient ion monitor (AIM, URG 9000D, URG Corporation, USA) (Shi et al., 2017), OC/EC analyzer (OCEC-100, Focused Photonics Inc, China) and a continuous atmospheric heavy metals monitoring system (AMMS-100, Focused Photonics Inc, China) (Ye et al., 2012), respectively. Twenty chemical species at 1 h time resolution were selected for analysis, including $NH_4^+$, $Na^+$, $Mg^{2+}$, $K^+$, $Cl^-$, $NO_3^-$, $SO_4^{2-}$, As, Ca, Tl, Br, Cs, Pb, Se, Cr, Zn, Fe, Mn, OC, and EC. Other species were excluded since they contained more than 40% values below the detection limit.

Each TS component contribution to the total variance of $PM_{2.5}$ and the chemical species concentrations in Tianjin was listed in Table S7. For $PM_{2.5}$, OC, EC, and inorganic ions (except $Mg^{2+}$ and $Na^+$), the diurnal and synoptic TS components had higher relative contributions to the total variance of these species than the intra-day and baseline TS components. Baseline TS component had largest relative contributions to the total variance of Ca, followed by the intra-day, intra-day, and diurnal TS components. For other elements (except Tl and Br), diurnal and synoptic/intra-day TS components had the larger amplitudes and were the larger contributors to the total variance of the concentrations. Dataset observed in Tianjin was processed following the 2.4 section to create RI, RD, RS, RBL datasets. For $PM_{2.5}$ and all chemical species, the largest average concentrations decline occurred when removed baseline TS component from original dataset (Table S8), suggesting baseline TS components dominating the average concentrations of $PM_{2.5}$ and chemical species.

The original, RI, RD, and RS datasets were respectively introduced into ME-2 to identify the sources of $PM_{2.5}$ (Figures 5 and 6). The regression analysis for the modeled and measured species mass concentrations shown that the slopes and r values of $PM_{2.5}$

were ranged from 0.65 to 0.87 and 0.80 to 0.94, respectively (Figure S13). Performance of solutions from RI and BL datasets are better than the solution from the original dataset, due to slops and r values were closer to 1 than the corresponding results from the original dataset. Comparable performance was obtained from RD, RS, and Original datasets. Six source categories were identified using ME-2 from both original and RD datasets, including crustal dust, vehicle emissions, coal combustion, secondary formation, biomass burning & sea salt, and industrial source. Biomass burning & sea salt was characterized by $K^+$ and $Cl^-$ (Tian et al., 2018; Zhu et al., 2018). The industrial source has high loadings of Zn (Monsalve et al., 2018; Ojekunle et al., 2018). While five sources were obtained from both RD and RS dataset except industrial source that mixed with crustal dust. Crustal dust & industrial source was identified by Ca, Fe, and Zn. We tried to separate the crustal dust and indusial source by adding factor number. However, the adding factor did not has a noticeable characteristic of factor profile and cannot be explained by crustal dust or indusial source. For Tianjin dataset, crustal dust was identified and mixed with the industrial source for the RD and RS datasets, suggesting removing diurnal or synoptic TS component affects source identification. Nitrate and sulfate did not separate from each other after removing the synoptic TS component, because they have similar variation trend. The correlation was 0.70 for the RS dataset, which is close to the results of other datasets (0.80, 0.81, and 0.84 in the original, RI, and RD datasets, respectively).

The correlation coefficient of the time series of source impacts between the original and the RI, RD, and RS datasets was listed in Table S9. Synoptic TS component

mainly influences secondary formation impact variation due to a relative low correlation of temporal trend between original and RS datasets. Other four sources, including crustal dust, vehicle emissions, coal combustion, and biomass burning & sea salt, are mainly affected by diurnal and synoptic scale influences. The average impacts of individual source categories on $PM_{2.5}$ from the datasets with removed TS components (Table 4). Vehicle emissions, crustal dust, coal combustion, biomass burning & sea salt, and indusial source were combined for the TPS. For the entire sampling period, the impacts of TPS obtained from the original, RI, RD, and RS datasets were similar to each other, ranging from 33.8 to 38.2 $\mu g$ $m^{-3}$. The TSS (secondary formation) solutions from the original and RI datasets exhibited similar source impacts, accounting for about 33 $\mu g$ $m^{-3}$, which was higher than the solution from the RD (27.3 $\mu g$ $m^{-3}$) and RS dataset (23.0 $\mu g$ $m^{-3}$). The RBL dataset, including about 45% negative values, was analyzed by PCA (Table S10). Four factors, including crustal dust (44.9%), secondary formation (8.7%), industrial source & coal combustion (6.4%), and vehicle (5.4%), were extracted and accounted for 65.4% of the total variance. In additional, ME-2 was applied to the baseline dataset and identified crustal dust, vehicle emissions, coal combustion, and secondary formation (Figure S14 and Table S11). The average TPS and TSS impacts on $PM_{2.5}$ mass concentrations were 29.6 $\mu g$ $m^{-3}$ (58%) and 21.3 $\mu g$ $m^{-3}$ (42%) respectively.

Seasonal variations of source impacts were investigated in Tianjin, as shown in Table 4. Winter period in this work included 15 days in March 2018 because 15 days is too short to present spring. The TPS impacts derived from the original, RI, RD, and RS

datasets were relatively stable, ranging from 31.9 to 35.8μg m⁻³ during fall and 33.6 to 37.1 μg m⁻³ during winter (Table 4). The TSS impacts decreased from 27.6 (original dataset) to 20.5μg m⁻³ (RS dataset) during the fall and from 39.0 (original dataset) to 26.6μg m⁻³ (RS dataset) during winter. A more significant decrease in secondary source impact during winter suggested that the synoptic TS component had a more significant impact on the secondary source during winter than during the fall in this work.

[Figure]

Figure 5 The influence of different TS components on source determination (Tianjin site). The industrial source was not identified from the RD and RS datasets. RBL dataset was investigated by PCA analysis instead of ME-2 due to the dataset has some negative values.

[Figure]

Figure 6 Source contributions to PM₂.₅ for each source (vertical columns) and each dataset (horizontal rows) (Tianjin site). The blanks mean that the source has not been identified.

[Figure]

**Figure S13.** The performance of ME-2 from five datasets (Tianjin site). (Left): the slops between model and measured concentrations of chemical species and PM₂.₅. (RI: intra-day removed dataset, RD: diurnal removed dataset, RS: synoptic removed dataset, BL: baseline dataset) (Right)The correlation coefficients between modeled and measured concentrations of chemical species and PM₂.₅.

[Figure]

**Figure S14.** The factor profiles obtained from ME-2 from baseline dataset (Tianjin site).

**Table 4.** Average source contributions to PM$_{2.5}$ (μg m$^{-3}$) estimated by ME-2 from Tianjin for the original, RI, RD, and RS datasets during the entire sampling period.

| | | Crustal dust | Vehicle emission | Coal combustion | Biomass burning & sea salt | Industrial source | TPS[a] | Secondary formation (TSS[b]) |
|---|---|---|---|---|---|---|---|---|
| During the entire sampling period | Original | 4.3[c] (6%) | 10.2 (14%) | 8.5 (12%) | 5.0 (7%) | 10.2 (14%) | **38.2 (54%)** | **32.5 (46%)** |
| | RI | 3.8 (6%) | 9.2 (13%) | 8.9 (13%) | 5.9 (9%) | 8.6 (12%) | **36.5 (53%)** | **32.6 (47%)** |
| | RD | 9 (14%) | 10.9 (17%) | 12.8 (20%) | 3.5 (6%) | | **36.2 (57%)** | **27.3 (43%)** |
| | RS | 9.2 (16%) | 8.0 (14%) | 11.5 (20%) | 5.1 (9%) | | **33.8 (59%)** | **23.0 (41%)** |
| Fall[d] | Original | 3.7 (6%) | 8.7 (14%) | 10.4 (16%) | 3.0 (5%) | 10.0 (16%) | 35.8 (56%) | 27.6 (44%) |
| | RI | 2.8 (5%) | 7.4 (12%) | 11.2 (18%) | 3.2 (5%) | 8.9 (14%) | 33.5 (55%) | 27.9 (45%) |
| | RD | 7.9 (14%) | 6.4 (11%) | 15.2 (27%) | 2.9 (5%) | | 32.4 (58%) | 23.7 (42%) |
| | RS | 8.0 (15%) | 5.6 (11%) | 14.6 (28%) | 3.7 (7%) | | 31.9 (61%) | 20.5 (39%) |
| Winter | Original | 4.1 (5%) | 10.7 (14%) | 7.6 (10%) | 5.2 (7%) | 9.5 (12%) | 37.1 (49%) | 39.0 (51%) |
| | RI | 3.9 (5%) | 10.2 (14%) | 7.6 (10%) | 6.3 (8%) | 8.1 (11%) | 36.0 (48%) | 39.4 (52%) |
| | RD | 8.9 (13%) | 13.0 (19%) | 11.6 (17%) | 3.7 (5%) | | 37.1 (53%) | 32.7 (47%) |
| | RS | 9.7 (16%) | 9.4 (16%) | 9.2 (15%) | 5.3 (9%) | | 33.6 (56%) | 26.6 (44%) |

[a]TPS is the total contributions of crustal dust, vehicle emissions, coal combustion, biomass burning & sea salt, and industrial source. [b]TSS is the total contributions of secondary formation and nitrate source. [c]The data in the parentheses and outside the parentheses are the absolute values of average source contribution (μg m$^{-3}$) and percentages of average source contribution (%), respectively. [d] Fall included October and November, and winter included December, January, February, and March (15 days).

**Table S7.** Relative contributions (%) of the different TS components to the total variance of chemical species concentrations (Tianjin site).

| | Intra-day (%) | Diurnal (%) | Synoptic (%) | Baseline (%) |
|---|---|---|---|---|
| $PM_{2.5}$ | 6 | **25** | **60** | 9 |
| $Cl^-$ | 12 | **44** | **38** | 6 |
| $NO_3^-$ | 4 | **23** | **68** | 6 |
| $SO_4^{2-}$ | 4 | **25** | **59** | 12 |
| $NH_4^+$ | 3 | **18** | **62** | 17 |
| $Mg^{2+}$ | 9 | 24 | **40** | **27** |
| $K^+$ | 14 | **32** | **44** | 10 |
| $Na^+$ | 6 | 9 | **17** | **69** |
| OC | 14 | **34** | **48** | 4 |
| EC | 10 | **33** | **49** | 8 |
| As | **34** | **43** | 21 | 2 |
| Ca | **24** | 20 | **24** | **33** |
| Tl | **32** | 7 | 2 | **59** |
| Br | **26** | **53** | 19 | 2 |
| Cs | 16 | 13 | **18** | **53** |
| Pb | 23 | **39** | **33** | 6 |
| Se | 28 | **34** | **30** | 8 |
| Cr | **48** | 29 | 14 | 9 |
| Zn | **26** | **44** | 23 | 6 |
| Fe | 27 | **31** | **29** | 13 |
| Mn | **29** | **33** | **29** | 9 |

**Table S8.** Average concentrations of $PM_{2.5}$ and chemical species for five datasets (Tianjin site).

| | Original | RI | RD | RS | RBL |
|---|---|---|---|---|---|
| $PM_{2.5}$($\mu$g m$^{-3}$) | 69.7 | 68.1 | 62.8 | 57.4 | 21.1 |
| $Cl^-$ ($\mu$g m$^{-3}$) | 3.8 | 3.6 | 3.1 | 3.1 | 1.7 |
| $NO_3^-$ ($\mu$g m$^{-3}$) | 14.3 | 13.9 | 12.1 | 9.5 | 7.5 |
| $SO_4^{2-}$($\mu$g m$^{-3}$) | 7.4 | 7.3 | 6.7 | 5.8 | 2.6 |
| $NH_4^+$ ($\mu$g m$^{-3}$) | 13.9 | 13.7 | 12.9 | 11.6 | 3.6 |
| $Mg^{2+}$($\mu$g m$^{-3}$) | 0.05 | 0.05 | 0.04 | 0.04 | 0.01 |
| $K^+$ ($\mu$g m$^{-3}$) | 0.9 | 0.9 | 0.8 | 0.8 | 0.3 |
| $Na^+$ ($\mu$g m$^{-3}$) | 0.9 | 0.9 | 0.9 | 0.9 | 0.05 |
| OC ($\mu$g m$^{-3}$) | 7.3 | 7.1 | 6.7 | 6.6 | 1.4 |
| EC ($\mu$g m$^{-3}$) | 3.9 | 3.7 | 3.4 | 3.2 | 1.5 |
| As (ng m$^{-3}$) | 5.7 | 4.5 | 4.4 | 4.5 | 3.8 |
| Ca (ng m$^{-3}$) | 312.8 | 289.0 | 283.4 | 281.8 | 84.5 |
| Tl (ng m$^{-3}$) | 2.8 | 2.1 | 2.6 | 2.7 | 0.9 |
| Br (ng m$^{-3}$) | 33.0 | 28.5 | 26.0 | 27.0 | 17.9 |
| Cs (ng m$^{-3}$) | 9.4 | 8.3 | 8.6 | 8.0 | 3.4 |
| Pb (ng m$^{-3}$) | 63.4 | 57.1 | 53.4 | 52.7 | 27.4 |
| Se (ng m$^{-3}$) | 6.1 | 5.0 | 4.8 | 4.8 | 3.8 |
| Cr (ng m$^{-3}$) | 6.1 | 4.7 | 4.7 | 5.2 | 3.8 |
| Zn (ng m$^{-3}$) | 270.7 | 235.4 | 211.5 | 222.9 | 145.1 |
| Fe (ng m$^{-3}$) | 600.8 | 545.5 | 528.6 | 538.4 | 191.0 |
| Mn (ng m$^{-3}$) | 42.9 | 36.3 | 33.6 | 33.2 | 26.1 |

**Table S9.** Correlation coefficients (3378 samples) between original and intra-day removed dataset, diurnal removed dataset, and synoptic removed dataset for source contributions (Tianjin site).

| | RI | RD | RS |
|---|---|---|---|
| Crustal dust[a] | 0.89** | 0.69** | 0.52** |
| Vehicle emission | 0.96** | 0.82** | 0.76** |
| Coal combustion | 0.88** | 0.68** | 0.59** |
| Secondary formation | 0.99** | 0.93** | 0.75** |
| Biomass burning & sea salt | 0.97** | 0.74** | 0.81** |
| Industrial source[b] | 0.90** | | |

[a]Crustal dust from RD and RS datasets mixed with industrial source. [b]Only ME-2 from original and RI datasets identified the Industrial source. **Significant correlation at 0.01 level.

**Table S10.** The results obtained from PCA from RBL dataset (Tianjin site).

| Components | Factor 1 | Factor 2 | Factor 3 | Factor 4 |
|---|---|---|---|---|
| $Cl^-$ | 0.33 | 0.20 | **0.78** | 0.23 |
| $NO_3^-$ | 0.26 | **0.82** | 0.10 | 0.27 |
| $SO_4^{2-}$ | 0.17 | **0.82** | 0.22 | 0.31 |
| $NH_4^+$ | 0.26 | **0.73** | 0.34 | 0.28 |
| $Mg^{2+}$ | 0.08 | 0.16 | -0.10 | **0.83** |
| $K^+$ | 0.21 | 0.26 | 0.32 | 0.77 |
| $Na^+$ | 0.28 | 0.23 | 0.21 | 0.17 |
| OC | 0.13 | 0.24 | 0.40 | **0.65** |
| EC | 0.24 | 0.20 | **0.61** | **0.51** |
| As | 0.10 | **0.62** | 0.41 | 0.07 |
| Ca | **0.75** | 0.06 | 0.21 | 0.04 |
| Tl | 0.02 | -0.17 | 0.04 | 0.04 |
| Br | 0.18 | 0.37 | **0.70** | 0.01 |
| Cs | 0.09 | 0.34 | 0.07 | -0.10 |
| Pb | 0.39 | 0.43 | **0.60** | 0.21 |
| Se | 0.47 | **0.57** | 0.30 | 0.11 |
| Cr | **0.73** | 0.20 | 0.07 | 0.23 |
| Zn | 0.47 | 0.18 | **0.71** | 0.07 |
| Fe | **0.85** | 0.21 | 0.34 | 0.08 |
| Mn | **0.80** | 0.29 | 0.28 | 0.12 |
| Variance contribution (%) | 44.9 | 8.7 | 6.4 | 5.4 |

**Table S11.** Average source contributions to $PM_{2.5}$ ($\mu g\ m^{-3}$) estimated by ME-2 from the BL datasets (Tianjin site).

| | Crustal dust | Vehicle emission | Coal combustion | TPS[a] | Secondary formation(TSS[b]) |
|---|---|---|---|---|---|
| During the entire sampling period | 7.3[c] (14%) | 12.0 (23%) | 10.4 (20%) | **29.6 (58%)** | **21.3 (42%)** |
| Fall | 3.4 (7%) | 12.8 (28%) | 16.5 (36%) | **32.7 (71%)** | **13.3 (29%)** |
| Winter[d] | 8 (15%) | 9.4 (17%) | 10.5 (19%) | **28 (51%)** | **26.4 (49%)** |

[a]TPS is the total contributions of crustal dust, vehicle emissions, and coal combustion. [b]TSS is the contributions of secondary formation. [c]The data in the parentheses and outside the parentheses are the absolute values of average source contribution ($\mu g\ m^{-3}$) and percentages of average source contribution (%), respectively. [d]The winter included 15 days in March 2018.

c) ME2 is a powerful tool to estimate the sources of PM and required model input

is non-negative. PCA was only used to analyze RBL dataset as negative values (about 40%) of the RBL dataset and the limitation of ME2.

PMF/ME2 does not need source profiles as model input but require receptor data. It is possible that one factor includes multiple sources and combine other chemical components in the factor profile except for the source markers (Canepari et al., 2009; Lee et al., 2009). Given that coal combustion, vehicle emissions, and crustal dust are collinearity sources that have similar profiles (Shi et al., 2009; Shi et al., 2011; Zhang et al., 2013), and that the absence of marker species (Si, Al, etc.) in the source profiles increased their collinear and uncertainties (Peng et al., 2016), these lead to difficulties in completely separating the three sources. We identified vehicle emissions and coal combustion according to the criterion that OC and EC fraction in vehicle emissions are higher than the values in coal combustion. We also analyzed the correlations of time series between source contributions and gaseous pollutants ($SO_2$, NO, $NO_2$) and diurnal patterns of source contribution (Figure S8). Correlation analysis showed vehicle emission has a significant correlation, with NO (ranging from 0.1 to 0.3, p<0.01) and $NO_2$ (ranging from 0.2 to 0.3, p<0.01). Correlation between coal combustion and $SO_2$ ranged from 0.4 (p<0.01) to 0.6 (p<0.01), for the four datasets. For the results of the original dataset, vehicle emissions exhibited a relatively high contribution to $PM_{2.5}$ during the nighttime, suggesting that diesel vehicles appeared and emitted pollutants during the nighttime (Gao et al., 2016). Coal combustion showed a stable diurnal trend, and crustal dust has high contributions from 0:00 to 12:00. The diurnal pattern of secondary formation primarily dominated by nitrate that peaked in the early morning

and at nighttime (Xu et al., 2014). Source diurnal trends estimated from RI dataset similar with results from original dataset, implying the small influences of intra-day TS component on the source diurnal trends. For the results from RS dataset, secondary formation (sulfate source) presented a broad peak during the daytime and might link to photochemical processes of sulfate. For results from RD and BL datasets, it is an expected result that all of four sources did not show obvious diurnal trends after removing the diurnal TS component.

[Figure]

**Figure S8.** Diurnal trend of Source impacts for each source (vertical columns) and each dataset (horizontal rows) (Beijing site). The blanks mean that the source has not been identified.

**References:**

Canepari, S., Pietrodangelo, A., Perrino, C., Astolfi, M. L., Marzo, M. L.: Enhancement of source traceability of atmospheric PM by elemental chemical Fractionation. Atmos. Environ. 43, 4754-4765, doi: 10.1016/j.atmosenv.2008.09.059, 2009.

Gao, J., Peng, X., Chen, G., Xu, J., Shi, G. L., Zhang, Y. C., and Feng, Y. C.: Insights into the chemical characterization and sources of $PM_{2.5}$ in Beijing at a 1-h time resolution, Sci. Total. Environ., 542, 162-171, doi:

10.1016/j.scitotenv.2015.10.082, 2016.

Lee, D., Balachandran, S., Pachon, J., Shankaran, R., Lee, S., Mulholland, J. A., and Russell, A. G.: Ensemble-trained PM$_{2.5}$ source apportionment approach for health studies. Environ. Sci. Technol., 43(18), 7023-7031, doi: 10.1021/es9004703, 2009.

Peng, X., Shi, G. L., Gao, J., Liu, J. Y., Huangfu, Y. Q., Ma, T., Wang, H. T., Zhang, Y. C., Wang, H., Li, H., Ivey, C. E., and Feng, Y. C.: Characteristics and sensitivity analysis of multiple-time-resolved source patterns of PM$_{2.5}$, with real time data using Multilinear Engine 2, Atmos. Environ., 139, 113-121, doi: 10.1016/j.atmosenv.2016.05.032, 2016.

Shi, G. L., Xu, J., Peng, X., Xiao, Z. M., Chen, K., Tian, Y. Z., Guan, X. B., Feng, Y. C., Yu, H. F., Nenes, A., and Russell, A. G.: pH of aerosols in a polluted atmosphere: source contributions to highly acidic aerosol. Environ. Sci. Technol., 51, doi: 4289-4296, 10.1021/acs.est.6b05736, 2017.

Shi, G. L., Xu, J., Peng, X., Xiao, Z. M., Chen, K., Tian, Y. Z., Guan, X. B., Feng, Y. C., Yu, H. F., Nenes, A., and Russell, A. G.: pH of aerosols in a polluted atmosphere: source contributions to highly acidic aerosol. Environ. Sci. Technol., 51, doi: 4289-4296, 10.1021/acs.est.6b05736, 2017.

Shi, G. L., Zeng, F., Li, X., Feng, Y. C., Wang, Y. Q., Liu, G. X., and Zhu, T.: Estimated contributions and uncertainties of PCA/MLR–CMB results: Source apportionment for synthetic and ambient datasets. Atmos. Environ. 45, 2811-2819, doi: 10.1016/j.atmosenv.2011.03.007, 2011.

Monsalve, S. M., Martínez, L., Vásquez, K. Y., Orellana, S. A., Vergara, J. K., Mateo, M. M., Salazar, R. C., Salazar, R. C., and Lillo, D. D. C.: Trace element contents in fine particulate matter (PM$_{2.5}$) in urban school microenvironments near a contaminated beach with mine tailings, Chañaral, Chile. Environ. Geochem. Health., 40, 1077-1091, doi: 10.1007/s10653-017-9980-z, 2018.

Ojekunle, Z. O., Jinadu, O. O. E., Afolabi, T. A., and Taiwo, A. M.: Environmental Pollution and Related Hazards at Agbara Industrial Area, Ogun State. Sci. Rep., 8. doi :10.1038/s41598-018-24810-4, 2018.

Tian, Y. Z., Liu, J. Y., Han, S. Q., Shi, X. R., Shi, G. L., Xu, H., Yu, H. F., Zhang, Y. F., Feng, Y. C., and Russell, A. G.: Spatial, seasonal and diurnal patterns in physicochemical characteristics and sources of $PM_{2.5}$ in both inland and coastal regions within a megacity in China. J. Hazard. Mater., 342, 139-149, doi: 10.1016/j.jhazmat.2017.08.015, 2018.

Xu, J., Zhang, Q., Chen, M., Ge, X., Ren, J., and Qin, D.: Chemical composition, sources, processes of urban aerosols during summertime in northwest China, insights from high-resolution aerosol mass spectrometry. Atmos. Chem. Phys., 2014;14:12593-12611, doi.org/10.5194/acp-14-12593-2014, 2014.

Ye, H. J., Liao, X. F., Guo, S. L., Jiang, X. J., Yao, L., and Chen, X. S.: Development and Application of Continuous Atmospheric Heavy Metals Monitoring System Based on X-Ray Fluorescence, Adv. Mater., Res., 518-523, 1510-1515, doi: 10.4028/www.scientific.net/AMR.518-523.1510, 2012.

Zhang, J. Q., Peng, L., Bai, H. L., Liu, X. F., and Mu, L.: Source Apportionment of Particulate Matter Based on Carbon Isotope Mass Balance Model. Appl. Mech. Mater. 295-298, 1565-1569, doi: org/10.4028/www.scientific.net/AMM.295-298.1565, 2013.

Zhu, Y., Yang, L., Chen, J., Kawamura, K., Sato, M., Tilgner, A., van Pinxteren, D., Chen, Y., Xue, L., Wang, X., Simpson, I. J., Herrmann, H., Blake, D. R., and Wang, W.: Molecular distributions of dicarboxylic acids, oxocarboxylic acids and α-dicarbonyls in $PM_{2.5}$ collected at the top of Mt. Tai, North China, during the wheat burning season of 2014, Atmos. Chem. Phys., 18, 10741-10758, https://doi.org/10.5194/acp-18-10741-2018, 2018.

---

## Author Comment (AC3) · 5 Sep 2018

We want to thank the reviewers for their support of the paper and their helpful comments and suggestions which have helped to improve our manuscript. We provide a point by point response to review.

**Response to Anonymous Referee #3**

The authors have proposed to improve the identification of major sources of ambient PM2.5 by conducting source apportionment upon the spectral decomposition of hourly measurements of ambient PM2.5 and its major ionic and trace element components. I think the manuscript falls short of fulfilling the above purposes. Several issues arise after reviewing the manuscript:

a) The process of decomposing the original data in time series of different temporal variations does not ensure that each of them is strictly positive, as they should. This is problematic for the subsequent application of ME2 (or PMF).

Response: We thank the reviewer for the comments and identifying some points that need to be addressed to strengthen the paper.

The temporal-scale components might show a positive or negative impact on pollutant concentrations resulting in some negative variations of them (Figures 1 and S6). So we designed an approach that can help solve more negative values in model input data than simply inputting time series of different temporal. We singly removed one a specific temporal-scale component (TS component) from the original dataset to ensure the input data are positive as much as possible, as shown in Figure. S5.